# Antioxidant Activities of a New Chemotype of *Piper cubeba* L. Fruit Essential Oil (Methyleugenol/Eugenol): In Silico Molecular Docking and ADMET Studies

**DOI:** 10.3390/plants9111534

**Published:** 2020-11-10

**Authors:** Fahad Alminderej, Sana Bakari, Tariq I. Almundarij, Mejdi Snoussi, Kaïss Aouadi, Adel Kadri

**Affiliations:** 1Department of Chemistry, College of Science, Qassim University, Buraidah 51452, Saudi Arabia; f.alminderej@qu.edu.sa (F.A.); k.aouadi@qu.edu.sa (K.A.); 2Faculty of Science of Sfax, Department of Chemistry, University of Sfax, B.P. 1171, Sfax 3000, Tunisia; sana.bakari@yahoo.fr; 3Department of Veterinary Medicine, College of Agriculture and Veterinary Medicine, Qassim University, P.O. Box 6622, Buraidah 51452, Saudi Arabia; tmndrj@qu.edu.sa; 4Department of Biology, Hail University, College of Science, P.O. Box 2440, Ha’il 2440, Saudi Arabia; snmejdi@gmail.com; 5Laboratory of Genetics, Biodiversity and Valorization of Bio-Resources (LR11ES41), University of Monastir, Higher Institute of Biotechnology of Monastir, Avenue Tahar Haddad, BP74, Monastir 5000, Tunisia; 6Faculty of Sciences of Monastir, University of Monastir, Avenue of the Environment, Monastir 5019, Tunisia; 7Faculty of Science and Arts in Baljurashi, Albaha University, P.O. Box (1988), Albaha 65527, Saudi Arabia

**Keywords:** *Piper cubeba* L., volatile oil, β-carotene assay, DPPH assay, in silico computational study, human peroxiredoxin 5

## Abstract

*Piper cubeba* L. fruit is an important species used in folk medicine for different types of pains such as rheumatism, chills, flu, colds, muscular aches, and fever. This study examines the chemical constituents, antioxidant activity, and potential inhibitory effect against human peroxiredoxin 5, a key enzyme of *P. cubeba* essential oil from fruits. Using gas chromatography coupled with mass spectrometry (GC–MS), the principal components were methyleugenol (41.31%) and eugenol (33.95%), followed by (E)-caryophyllene (5.65%), *p*-cymene-8-ol (3.50%), 1,8-cineole (2.94%), and α-terpinolene (1.41%). Results showed similar scavenging activity via 2,2-diphenyl-1-picrylhydrazyl DPPH radical scavenging activity (IC_50_ = 110.00 ± 0.08 μg/mL), as well as very potent antioxidant activity against both ferric reducing/antioxidant power (FRAP) (106.00 ± 0.11 μg/mL) and β-carotene bleaching (IC_50_ = 315.00 ± 2.08 μg/mL) assays when compared to positive butylated hydroxytoluene and ascorbic acid. The molecular docking approach has also been performed to screen the antioxidant activities of the major and potent compounds against human protein target peroxiredoxin 5. Results showed good binding profiles and attributed the strongest inhibitory activity to β-caryophyllene oxide (–5.8 kcal/mol), followed respectively by isocembrol and α-selinene (–5.4 kcal/mol), and viridiflorol (–5.1 kcal/mol). Furthermore, ADME (absorption, distribution, metabolism and excretion)-related physicochemical and pharmacokinetic properties have been assessed and support our in vitro findings. This work demonstrates the powerful antioxidant potency of cubeba pepper and paves the way for the discovery and development of antioxidant agent with high potency.

## 1. Introduction

Oxidative stress (OS), occurring through cellular damage as the major cause of defective generative function, arises from an imbalance of cell redox reactions, which is due to the formation of reactive oxygen/nitrogen species (ROS and RNS, respectively), such as superoxide anion (O_2_^–^•), peroxide radical (•OOH), singlet oxygen (^1^O_2_), hydroxyl radical (•OH), nitric oxide (NO•), and peroxynitrite (ONOO^–^). These reactions occur in human cells because of endogenous and exogenous factors, and are responsible of several human-related diseases [1,2,3,4,5]. OS is involved in several acute and chronic pathogeneses of aging and degenerative diseases, such as atherosclerosis, cardiovascular disease, diabetes, and cancer [5]. It has recently been shown that ROS, as secondary messengers, are the mainstay in cellular signaling and participate in protein synthesis, cell proliferation, cell survival, differentiation, inflammation, and glucose metabolism [6,7]. Antioxidants are compounds that can delay and inhibit or prevent the oxidation of oxidizing substances by scavenging free radicals and reducing oxidative stress, as these are the main cause of the development of chronic degenerative diseases, such as coronary heart disease, cancer, and aging. As plant-based natural products, essential oils and their bioactive molecules are a starting point for the development of lead compounds, and have been approved therapeutically as drugs for their potential antioxidant capacities. These can be attributed to the presence of terpenes, as well as the phenolic compounds that significantly enhance free radical scavenging activity, and therefore can substitute for synthetic antioxidants [8,9,10,11]. They possess versatile applications in cosmetics, food preservation, beverages, and fragrances, as well as multidimensional and multivariate biological properties, mainly anti-inflammatory, antimicrobial, antioxidant, anticancer, and antiviral powers, as well as the prevention of cardiovascular and degenerative diseases [12,13,14,15,16]. Indeed, drugs obtained from plants, such as essential oils, are generally acclaimed for their effectiveness, simple accessibility, and their lack of side effects. Their attractive effects are continuously growing, due to their usefulness for culinary, medicinal, and other anthropogenic applications.

The species *P. cubeba* belongs to the Piperaceae family, commonly known as cubeba in Arabic and tailed piper in English. This plant grows wild in Java and the Sumatra islands (Indonesia), as well as in southern India, and also grows wild in many other countries of Southeast Asia and in some African countries. The fruit of this plant is well-known for its medicinal properties [17,18]. In fact, cubeba pepper has been extensively used as a food additive (spices), and as a potent therapeutic agent in folkloric medicine against antimicrobial infection, bactericidal (*Helicobacter pylori*), renal, acute jaundice, diarrhea, syphilis, dysentery, asthma gonorrhea, abdominal pain, and enteritis as well as renal [19,20,21,22]. This work was conducted due to the lack of studies on the chemical composition of the *P. cubeba* fruit’s essential oil, in parallel with the search for new powerful chemotype, as well as the originality of this study concerning the correlation between in vitro evaluation of the antioxidant activity of *P. cubeba* essential oil from fruits with an in silico approach (molecular docking) targeting protein involved in oxidative stress (human peroxiredoxin 5).

Taking into consideration the above facts, the aim of the herein presented research was to explore the volatile chemical constituents and evaluate the in vitro antioxidant activity of *P. cubeba* fruit’s essential oil. In parallel, in silico prediction (ADMET (absorption, distribution, metabolism and excretion)) and molecular docking studies of the major chemical constituents are also carried out to support and account for the experimental results.

## 2. Results

### 2.1. Essential Oil Characterization

The light-yellow *P. cubeba* fruit’s essential oil was obtained from the hydrodistillation process, with a yield of 1.0% (*v/w*). The characterization of the volatile compounds through gas chromatography and mass spectrometry (GC–MS) was identified by comparing the retention time on the HP–5MS column and the molecular weight with reference compounds in the NIST library, which revealed the presence of 24 components, representing 98.13% of the total amount (Table 1).

Methyleugenol (41.31%) and eugenol (33.95%) were identified as the two major constituents in this oil, and therefore can be classified as a methyleugenol/eugenol chemotype. The other identified components were present at a lower level, with (E)-caryophyllene (5.65%), *p*-cymene-8-ol (3.50%), 1,8-cineole (2.94%), and α-terpinolene (1.41%).

Phenylpropanoids represented the main characterized class (75.41%), followed by monoterpenes, including both hydrocarbonated (9.71%) and oxygenated (3.06%). Hydrocarbonated sesquiterpenes were one of the identified compounds, with a concentration of 1.69%, from the overall identified compounds, in addition to those that were minor oxygenated (8.26%). 

### 2.2. Antioxidant Activity

Three complementary assays (DPPH, ferric reducing/antioxidant power (FRAP), and *β*-carotene bleaching) were used for the evaluation of the antioxidant potential of *P. cubeba* fruit’s essential oil.

#### 2.2.1. DPPH Test

Significant (*p* < 0.05) antiradical activity was recorded *via* the DPPH test (IC_50_ = 110.00 ± 0.08 μg/mL), which was very close to that of the positive control, ascorbic acid (IC_50_ = 114.00 ± 0 70 μg/mL) (Table 2).

#### 2.2.2. FRAP Test

The results of the FRAP assay, expressed in terms of EC_50_ (Table 2), showed that *P. cubeba* fruit’s essential oil has a powerful capacity (*p* < 0.05) to reduce ferric ions to ferrous ions, with an EC_50_ of 106.00 ± 0.11 μg/mL, which is three times higher than ascorbic acid, used as a standard (330.00 ± 0.60 μg/mL).

#### 2.2.3. *β*-Carotene Bleaching Test

Results of *β*-carotene bleaching test (Table 2) indicated that the antioxidant capacity of cubeba pepper essential oil (IC_50_ = 315.00 ± 2.08 μg/mL) was higher (*p* < 0.05) than that of the used synthetic antioxidant, butylated hydroxytoluene (BHT) (IC_50_ = 930.00 ± 0.02 μg/mL).

### 2.3. Molecular Docking Study

In order to corroborate the experimental data and understand the possible mechanism behind the antioxidant activity, molecular docking studies were examined using human peroxiredoxin 5 enzyme (PDB: 1HD2), which has a broader activity against ROS and was mostly involved in stress protection mechanisms and in cell differentiation. The listed binding energies of the formed complex between identified *P. cubeba* fruit’s essential oil constituents and peroxiredoxin 5 (Table 3) were found to be in the range −5.8 to −3.7 kcal/mol. From our molecular docking results, it can be suggested that all ligands interact favorably with the target proteins.

#### 2.3.1. Protein–Ligand Complex Interactions

Herein, we will focus mainly on both of the major constituents, methyleugenol (−4.3 kcal/mol) and eugenol (−4.7 kcal/mol), representing about 75% of the whole *P. cubeba* fruit’s essential oil composition, as well as on those with the four top docking scores, which are β-caryophyllene oxide (−5.8 kcal/mol) followed respectively by isocembrol and α-selinene (−5.4 kcal/mol), and viridiflorol (−5.1 kcal/mol) (Figure 1).

Their interactions with the catalytic site of the investigated enzyme were analyzed (Table 4 and Figure 2).

From our molecular docking results, it was found that β-caryophyllene oxide was identified as the most potent inhibitor of the enzyme, with human peroxiredoxin 5 interacting favorably with its catalytic site via two H-bonding interactions with Gly46 (2.73) and Arg127 (6.35); five van der Waals interactions with Pro40, Thr44, Cys47, Ile119, and Thr147; and four Alkyl/Pi–Alkyl interactions with Pro45 (3.75), Leu116 (4.31) (4.68), and Phe120 (4.89). Isocembrol compound performed several hydrophobic interactions with the following residues of human peroxiredoxin 5 (1HD2): one H bond with Thr147 (2.61), five van der Waals interactions with Thr44, Gly46, Ile119, Arg127, and Gly148l and five Alkyl/Pi–Alkyl interactions with Pro40 (4.03), Pro45 (3.87) (4.31), Leu116 (4.50), and Phe120 (5.31) amino acid residues. The interactions involved between α-selinene and human peroxiredoxin 5 are van der Waals interactions with Ala42, Thr44, Asn76, Phe120, and Arg124; Pi-Sigma interactions with Phe43 (3.80); and Alkyl/Pi–Alkyl interactions with Phe43 (5.26) and Val80 (4.19) (4.56). Thr147 (2.86) formed one H bond with a viridiflorol OH group. Other forms of interactions that were established included van der Waals reactions with Gly46, Cys47, Phe120, and Arg12; C–H bonds with Thr147 (3.28); and Alkyl interactions with Pro45 (4.75), Leu116 (4.13) (4.42) (5.43), and Ile119 (5.10) residues.

#### 2.3.2. ADME Analysis

As one of the best filters in the virtual screening of bioactive molecules, in order to be an effective drug in early preclinical development, the forecasting of ADME (absorption, distribution, metabolism and excretion) profiles of the selected compounds, including their pharmacokinetic and drug-like properties, have been investigated using Swiss ADME (http://www.swissadme.ch/). The selected phytocompounds (Table 5) were found to correctly meet the Lipinski Rule of five, and also share topological polar surface area TPSA values less than 30 Å^2^, suggesting good brain penetration and good lipophilicity behavior, which is expressed by the consensus Log *P*_o/w_ in the range 2.25–4.75. Their bioavailability score of 0.55 indicates their more drug-like properties. Noticeably, there is no P-glycoprotein (P–gp) substrate justifying their good intestinal absorption and bioavailability; the compounds exhibited high gastrointestinal absorption (GI) (except α-selinene), and only isocembrol and α-selinene were predicted to not cross the blood–brain barrier (BBB). However, the others easily pass the blood–brain barrier (BBB) permeant, and can bind to specific receptors. The study’s components interacted at most with two isoenzymes of the Cytochrome P (CYP) family, confirming their better effectiveness with insignificant toxicity.

Their radar plot (Figure 3A) indicates that all selected compounds are entirely inside the pink area, indicating their better drug likeness alone with a good bioavailability profile. Additionally, pharmacokinetics were evaluated by the boiled-egg model (Figure 3B). The compounds methyleugenol, eugenol, β-caryophyllene, and viridiflorol appeared in the yellow region (yolk) with a red point, which confirms their high probability of brain penetration acting as a non-substrate of P–gp (PGP−).

## 3. Discussion

### 3.1. Chemical Composition Analysis vs. Beneficial Role

Plants used as condiments have been noted for their extremely beneficial therapeutics, and have been described with a wide variety of antioxidant compounds able to prevent oxidative stress via their complex composition and richness in bioactive molecules.

As shown, for this oil, phenylpropanoids form the backbone of the characterized compounds. The richness of *P. cubeba* fruit’s essential oil on phenylpropanoids, which are chemically a group of compounds derived from the carbon skeleton of phenylalanine, implicated in plant defense, structural support, and the survival of plants, have been widely reported for their anticonvulsant, analgesic, antidiabetic, and anti-inflammatory potential [24]. Furthermore, they have shown application in the food, pharmaceuticals, cosmetics, textiles, biofuel, and sensors [24]. Macchia et al. [25] have indicated that methyl eugenol is formed through cyclic acid pathways, with phenylalanine as precursor; however, eugenol will be formed under a few reaction stages. Eugenol is then converted easily to methyl eugenol via a methyl donor from sadenosylmethionine (SAM) and enzymatic activity from O-methyltransferase (OMT) [26]. Binu et al. [27] showed that treatment with eugenol improved the peroxidation of the membrane and restored normal heart rate. Also, histological examination of the cardiac segments has confirmed the beneficial role of eugenol against arsenic-induced oxidative damage. Eugenol has been reported to possess a large spectrum of activity, including antitumor, anti-inflammatory, analgesic, antibacterial, antifungal, anesthetic, and antipyretic potency [28,29,30,31,32,33,34].

On the other hand, the chemical composition of *P. cubeba* fruit’s essential oil has been scarcely explored. Previous reports have shown that its main constituents are β-elemene (9.4%) and sabinene (9.1%) [35]. In another study by Burfield [36], β-caryophyllene, δ-cadinene, α- and β-cubebene, and minor amounts of monoterpenes were found as the main phytocompounds. Lawrence [37] reported that the main components were α-copaene, β-cubebene, allo-aromadendrene, γ-muurolene, and germacrene D, followed by δ-cadinene and β-caryophyllene. In the recent study of Andriana et al. [38], the authors found that terpinen-4-ol (42.41%), α-copaene (20.04%), and γ-elemene (17.68%) were the major components. Based on the above literature descriptions, our chemical composition results are still different, and confirm the originality of our *P. cubeba* fruit’s essential oil.

### 3.2. Antioxidant Potential

The results prove that *P. cubeba* fruit’s essential oil is able to act as a donor of hydrogen atoms or electrons to reduce the DPPH radical. The potent DPPH scavenging activity of *P. cubeba* fruit’s essential oil could be attributed to the high amount of eugenol and methyleugenol [39,40]. Additionally, Gülçin in 2011 [41] showed that eugenol had the most powerful antioxidant activity and radical-scavenging activity compared to the standards: butylated hydroxyanisole, butylated hydroxytoluene, α-tocopherol, and Trolox. Recently, Gogoi et al. [39] reported that methyleugenol-rich lemongrass essential oil displayed the strongest antioxidant activity on DPPH assay, with IC_50_ = 2.263 μg/mL, which is lower than standard ascorbic acid (IC_50_ = 2.58 μg/mL). Also, other compounds in small amounts contribute to the scavenging activity, such as oxygenated sesquiterpenes [42].

Our FRAP results were confirmed by those mentioned previously by Zhang et al. [43], indicating the remarkable potency and contribution of eugenol (representing 33.95% in *P. cubeba* fruit’s essential oil via FRAP assay) to donate electrons to reactive free radicals, transforming them into more stable, non-reactive species, and terminating the free radical chain reaction.

β-carotene bleaching results indicate that this potent antioxidant capacity is probably attributable to the presence of diverse compounds in the tested volatile oil, such as terpinen-4-ol, 1,8-cineole, and *p*-cymene. In fact, Radonic et al. [44] and Tepe et al. [45] showed that oxygenated monoterpenes exert a strong inhibitory effect on the coupled oxidation of *β*-carotene/linoleic acid.

Indeed, based on the above results, the higher antioxidant activity of *P. cubeba* fruit’s essential oil could be related to this synergistic effect. The high content of methyleugenol/eugenol (75.26%) associated with other constituents (major and minor components) also could have been responsible for their bioactivities. Eugenol possesses the ability to transfer electron or hydrogen atoms by neutralizing free radicals, which can block the oxidative process [46,47]. The presence of methyleugenol/eugenol as the principal chemotype of this oil gives him better antioxidant potency, and therefore good DNA damage protective effects, which can be manifested via direct trapping of the free radicals or inhibiting the propagation of radical chain reactions through transfers of hydrogen or electrons. Overall, the significant antioxidant potential of *P. cubeba* fruit’s essential oil confirms the uses of cubeba fruit as a source of antioxidants, which may provide health-promoting advantages to consumers.

### 3.3. In Silico Study

Computational methods associated with experimental strategies have been of great value in modern drug design, and in the development and discovery of novel promising compounds. Our docking results were also well-correlated with in vitro antioxidant assays, along with the selected phytocompounds interacting perfectly with the protein target. Indeed, in addition to methyleugenol and eugenol, which are discussed above, the natural bicyclic sesquiterpene β-caryophyllene oxide has been proven to have potent antioxidant activity [48]. Isocembrol, also known as thunbergol, is an oxygenated diterpene (alcohol) detected in soft coral eggs *Lobophytum compactum* and *Lobophytum crissum*, and has been reported to have potential antioxidant activity [49,50]. Mayachiew and Devahastin [51], using a β-carotene/linoleic acid assay, reported that the *Alpinia galangal* extract rich α-selinene (called also α-Eudesma-3,11-diene) possesses potent antioxidant activity. Viridiflorol, as an oxygenated sesquiterpenes, was found to possess antioxidant properties [52]. The synergistic action of other minor components contributes to an increase in antioxidant activity that may be taken into account.

As stated above from the docking results, we found that they are in perfect agreement with those obtained by Declercq et al. [53] when analyzing the crystal structure of human peroxiredoxin 5 (1HD2). The authors demonstrated that at the active site of the enzyme, the residue Cys47 located at the N-terminal part of the kinked helix α2, inside a small cavity, was directly involved in peroxide reductase activity by forming an intramolecular disulfide intermediate in the oxidized enzyme. Also, the amino acid Arg127, interacting with the sulfur atom of Cys47 at a distance of 3.3 Å, seems to be responsible for the positively charged active site pocket. On the other hand, Thr44 is present in the active site cavity of human peroxiredoxin 5, with its oxygen atom clearly interacting with the sulfur atom of the catalytic cysteine residue (Cys47) at a distance of 3.0 Å. In addition, they found that one side of the active site pocket contains several hydrophobic residues, including Leu116, Ile119, and Phe120, whose side-chains are located near the benzoate aromatic ring, which is shown to act as hydroxyl radical scavenger (via its benzoate ion). As shown, numerous interactions between human peroxiredoxin 5-selected phytocompounds are close to the active site of the enzyme. Our finding also corroborates perfectly those of Eze et al. [54], demonstrating that the docked 1-(Phenylsulphonyl)-N-propylpyrrolidine-2-carboxamide with human peroxiredoxin 5 exhibited higher binding energy (−13.86 kcal mol^−1^), creating interactions with Thr44, Pro40, Pro 45, Gly46, Arg127, Thr147, and Cys47 residues. Additionally, with the same therapeutic target, caryophyllene oxide (−7.2 kcal/mol) from *Cymbopogon citratus* essential oil established non-covalent interactions with Pro 40, Thr 147, Thr 44, Phe 120, Pro 45, Leu 116, and Ile 119, sharing a large number of common residues with the active site of the investigated enzyme.

In silico ADME and drug-likeness prediction of the selected *P. cubeba* fruit essential oil’s bioactive molecules demonstrate good pharmacokinetic properties, which encourages further in vivo and in vitro evaluation study of the proposed essential oil to validate the computational findings.

## 4. Materials and Methods

### 4.1. Reagents, Plant Material, and Essential Oil Extraction

Reagents DPPH, ascorbic acid (AA), ferric chloride (FeCl_3_), butylated hydroxytoluene (BHT), potassium ferrocyanide (K_3_Fe(CN)_6_), ferrous sulfate (FeSO_4_), potassium persulfate (K_2_S_2_O_8_), Tween 60, sodium chloride, linoleic acid, β-carotene, and solvents used were all purchased from Sigma, France. Spectrophotometric measurements were performed using a double-beam UV-Vis spectrophotometer. 

Cubeba pepper fruits (Figure 4) were purchased commercially from the Saudi market (Baljurashi city, Albaha, Kingdom Saudi Arabia). Samples were cleaned well with tap water, sterilized, and dried in an oven at 60 °C for 72 h before being pulverized into a fine powder for possible use. Dried powder of the fruits (800 g) was subjected to hydrodistillation for 4 h using a Clevenger-type apparatus [51]. The resulting essential oils were dried using anhydrous sodium sulfate, and stored in sealed vials in the dark, at 4 °C for subsequent analysis.

### 4.2. Characterization of Essential Oils by GC–MS

The chemical composition of *P. cubeba* fruit’s essential oil has been identified using a GC–MS apparatus (Agilent Technologies, Santa Clara, CA, USA), as described by Bakari et al. [55]. The column used is an HP–5MS (5% phenyl) of length 30 m, inside diameter 0.25 mm, and film thickness 0.25 μm. The temperature of the injector and detector was 250 °C. The oven temperature was maintained at 100 °C for 1 min, then increased to 260 °C at a rate of 4 °C/min, and then maintained at 260 °C for 10 min. Helium gas has been used as a carrier gas at a constant rate of 1 mL/min. The injection volume was 1 μL. Ionization conditions by impact electronics (EI) were at an ionic energy of 70 eV and a range of swept mass of 50 to 550 *m*/*z*. 

### 4.3. In Vitro Antioxidant Activity

All tests were conducted in triplicate, and mean values were taken.

#### 4.3.1. DPPH (1,1-Diphenyl-2-picrylhydrazyl) Radical Scavenging activity

The DPPH test was carried out as described by Felhi et al. [56]. A series of dilutions was prepared from the *P. cubeba* fruit’s essential oil solution, each one supplemented with 1 mL of the ethanolic solution of DPPH. Then, the dilutions were incubated for 30 min in the dark, and the absorbance of the mixtures was measured at 517 nm against a negative control treated under the same conditions. Ascorbic acid (AA) was used in this test as the positive control.

#### 4.3.2. Ferric Reducing/Antioxidant Power (FRAP) Assay

The reducing power of the *P. cubeba* fruit’s essential oil was evaluated using the method described by Bakari et al. [9]. The sample was mixed with 2.5 mL of potassium phosphate buffer (0.2 M, pH 6.6) and 2.5 mL of potassium ferricyanide (1%). The mixture was incubated for 20 min at 50 °C in a water bath, and then 2.5 mL of trichloroacetic acid (10%) was added.

After centrifugation at 3000 rpm for 10 min at 25 °C, the supernatant was mixed with 2.5 mL of H_2_O and 0.5 mL of ferric chloride (0.1% FeCl_3_). A blank without a sample was prepared under the same conditions. The optical density was subsequently measured at 700 nm, and BHT (Butylated hydroxytoluene) was used as positive control.

#### 4.3.3. β-Carotene Bleaching Test

The β-carotene bleaching test was performed based on the experimental protocol done by Kadri et al. [8]. Briefly, a mixture containing 20 μL of linoleic acid and 200 mg of Tween 20% (emulsifier) was added to a previously prepared solution by dissolving 0.2 mg of β-carotene in 1 mL of chloroform. After vacuum evaporation of the chloroform at 40 °C, a volume of 50 mL of oxygen sparged water was added to the obtained emulsion. Then, a volume of 5 mL of this emulsion was introduced into a tube containing 0.5 mL of the sample or BHT (reference synthetic antioxidant). Control solutions have been prepared in parallel. The antioxidant activity was determined according to the optical density of the sample at 470 nm against a control solution prepared as above, but devoid of β-carotene.

### 4.4. Computational Approach

Interactions between the natural compounds and human peroxiredoxin 5 were assessed by in silico molecular docking, in order to explore the preferred orientation of the ligands in the binding site of receptors. The crystal structure was obtained from the Protein Data Bank: human peroxiredoxin 5 (PDB Code: 1HD2). All water molecules were removed, as was the co-crystallized ligand from the structures. Polar hydrogens and Gasteiger charges were assigned with AutoDockTools1.5.2 (ADT), and the PDBQT file format was prepared [57]. The same software was used to select a docking grid. In human peroxiredoxin 5 enzyme (PDB: 1HD2), the grid box site was established at 7.611, 43.828, and 29.921Å (*x*, *y*, and *z*, respectively) using a grid of 70, 58, and 58 points (*x*, *y*, and *z*, respectively) and including a spacing of 0.375 Å.

The structures of the natural compounds were minimized using a conjugate gradient AMMP incorporated in VEGA ZZ [58]. The conversion of the file from PDB to PDBQT was done using AutoDockTools1.5.2. We applied AutoDock Vina software with an exhaustiveness parameter of 32 to perform the docking simulations [59]. ADT was used for the docking conformation analysis. Receptor ligand interactions were visualized by Discovery Studio Visualizer (Dassault Systemes BIOVIA, 2015) [60].

### 4.5. ADMET Analysis

The pharmacokinetic and drug-like properties of the selected compounds were estimated using ADME (absorption, distribution, metabolism and excretion) descriptors by a SwissADME online server (http://www.swissadme.ch/), as previously reported [61,62,63].

### 4.6. Statistical Analysis

A one-way analysis of variance (ANOVA) and Tukey’s post-hoc test was performed to determine significant differences between the responses using the SPSS 19 statistical package (SPSS Ltd., Woking, UK). Means and standard deviations were calculated. Differences between the mean values of the various treatments were determined by the least significant difference test. A probability level of *p* < 0.05 was used in testing the statistical significance of all experimental data.

## 5. Conclusions

The current study aimed to explore a new *P. cubeba* fruit essential oil chemical composition and its antioxidant activity, in order to provide more information about its health benefits. The oil exhibited higher antioxidant activity than the tested standards. In silico prediction and molecular docking studies showed good correlation between the observed inhibitory activity and the in silico molecular docking scores, with methyleugenol, eugenol, β-caryophyllene oxide, isocembrol, α-selinene, and viridiflorol, the main contributors of the antioxidant activity, sharing many common residues with the human peroxiredoxin 5 active site. Therefore, they may be potent inhibitors, in synergism effect with minor compounds. Our findings demonstrate for the first time that *P. cubeba* fruit’s essential oil can be a new potential resource for future research to design new and more potent antioxidative agents.

## Figures and Tables

**Figure 1 plants-09-01534-f001:**
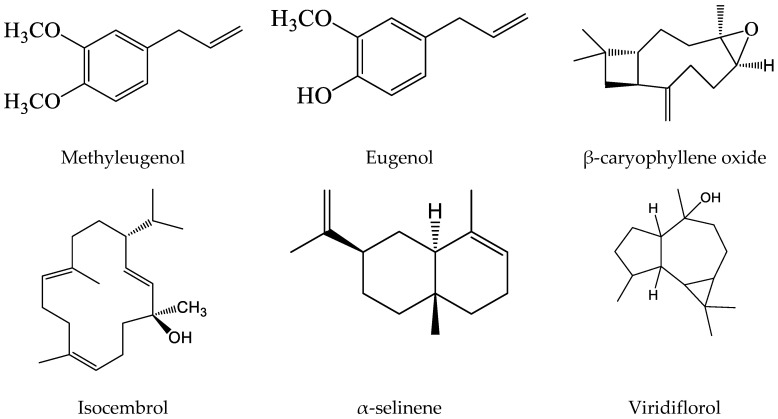
Chemical structure of the major volatile constituents, as well as those having the four top docking scores found in cubeba pepper fruit’s essential oil, as determined by gas chromatography coupled with mass spectrometry (GC–MS) technique.

**Figure 2 plants-09-01534-f002:**
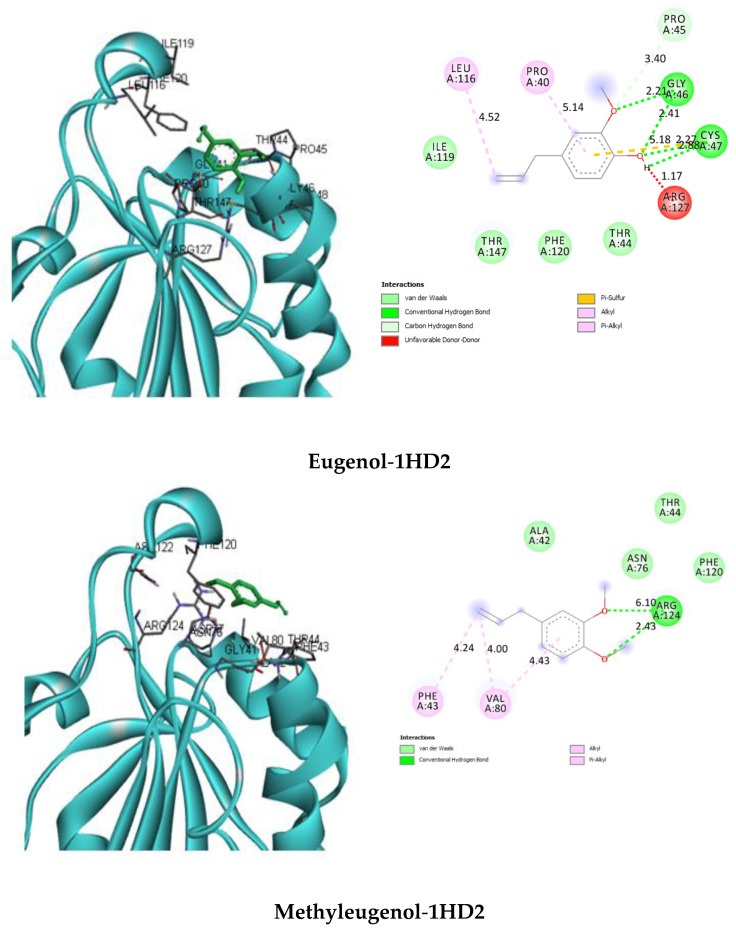
Interactions of human Peroxiredoxin 5 receptor (PDB: 1HD2) with the selected cubeba pepper fruit’s essential oil compounds.

**Figure 3 plants-09-01534-f003:**
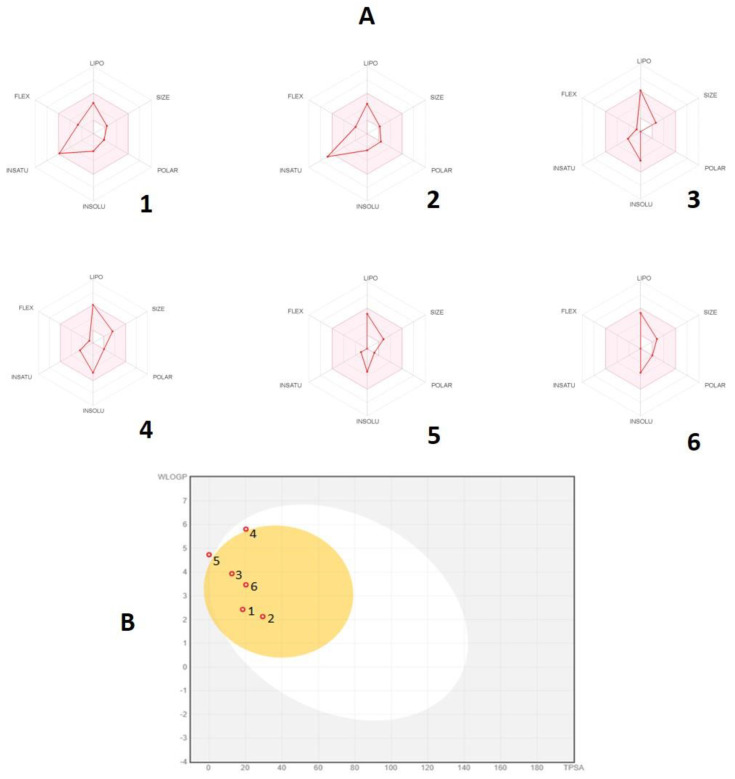
Bioavailability radar (**A**) and boiled-egg graph (**B**) of the selected phytoconstituants. 1: Methyleugenol, 2: eugenol, 3: β-caryophyllene oxide, 4: isocembrol, 5: α-selinene, 6: viridiflorol.

**Figure 4 plants-09-01534-f004:**
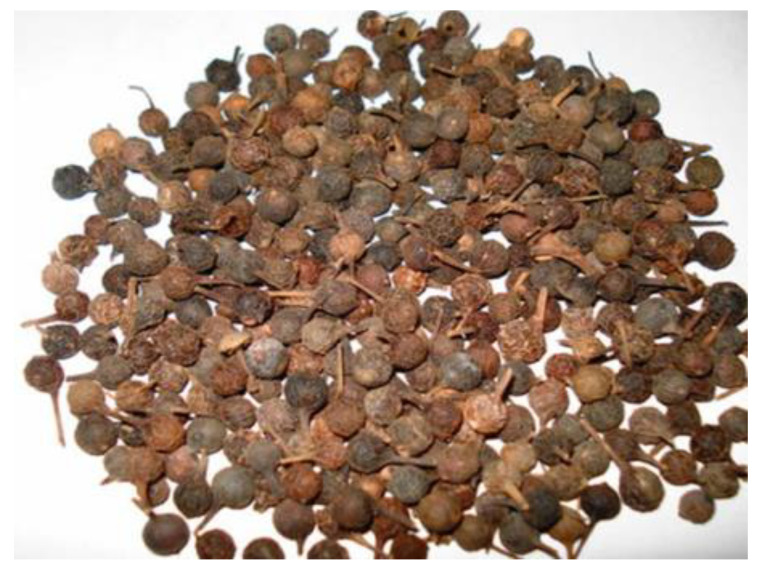
Cubeba pepper (*P. cubeba* L.) fruit.

**Table 1 plants-09-01534-t001:** Constituents present in *P. cubeba* fruit’s essential oil, as determined by gas chromatography and mass spectrometry (GC–MS), with methyleugenol/eugenol as new chemotype.

N°	Compounds	RI ^a^	RI ^b^	*R_t_* (mn)	% Area
1	β-myrcene	990	985	8.57	1.23
2	Limonene	1032	1031	9.52	0.12
3	1,8-cineole	1035	1039	9.68	2.94
4	β-ocimene	1049	1043	9.95	0.30
5	α-terpinolene	1089	1088	11.25	1.41
6	Linalool	1102	1098	12.88	0.22
7	Terpinen-4-ol	1177	1177	13.14	1.80
8	*p*-cymene-8-ol	1183	1183	13.47	3.50
9	α-terpineol	1186	1189	13.55	0.96
10	Estragole	1196	1195	14.14	0.15
11	Citronellol	1224	1228	14.19	0.10
12	(E)-geraniol	1248	1276	14.74	0.19
13	Eugenol (main compound 2)	1359	1356	17.30	33.95
14	β-elemene	1393	1391	17.73	0.66
15	Methyleugenol (main compound 1)	1404	1404	18.29	41.31
16	(E)-caryophyllene	1419	1418	18.45	5.65
17	α-humulene	1454	1444	19.06	1.14
18	Germacrene D	1486	1484	19.55	0.15
19	α-selinene	1490	1515	19.81	0.47
20	δ-cadinene	1526	1534	20.24	0.19
21	Spathulenol	1578	1573	21.37	0.18
22	β-caryophyllene oxide	1583	1580	21.52	0.96
23	Viridiflorol	1593	1591	22.72	0.39
24	Isocembrol	2044	2046	22.95	0.16
Total identified %					98.13
Phenylpropanoids					75.41
Oxygenated monoterpenes					9.71
Hydrocarbonated sesquiterpenes					8.26
Hydrocarbonated monoterpenes					3.06
Oxygenated sesquiterpenes					1.69

^a^ RI: literature retention indices on HP-5MS column, according to Adams, 2007 [23]. ^b^ RI: experimentally retention index calculated against a C8–C25 n-alkanes mixture on the HP-5MS column.

**Table 2 plants-09-01534-t002:** IC_50_ values of the DPPH free radical scavenging, FRAP, and *β*-carotene–linoleate assays of cubeba pepper fruit’s essential oil.

Samples	IC_50_ (μg/mL)		
	DPPH	FRAP	*β*-Carotene
Cubeba pepper EO	110.00 ± 0.08 ^a^	106.00 ± 0.11 ^a^	315.00 ± 2.08 ^a^
Butylated hydroxytoluene (BHT)	-	-	930.00 ± 0.02 ^b^
Ascorbic acid (AA)	114.00 ± 0.70 ^a^	330.00 ± 0.60 ^b^	-

± Standard deviation for the three replicates. EO: essential oil. Values followed by the same letter under the same column, for each parameter, are not significantly different (*p* < 0.05).

**Table 3 plants-09-01534-t003:** Binding energy of cubeba pepper fruit’s essential oil constituents, complexed with human peroxiredoxin 5 enzyme.

N°	Compounds	Receptor/Binding Energy (kcal/mole) (kcal/mol)
1	β-myrcene	−3.7
2	Limonene	−3.9
3	1,8-cineole	−4.3
4	β-ocimene	−4.0
5	α-terpinolene	−4.0
6	Linalool	−4.2
7	Terpinen-4-ol	−4.4
8	*p*-cymene-8-ol	−4.1
9	α-terpineol	−4.3
10	Estragole	−4.1
11	Citronellol	−4.0
12	(E)-geraniol	−3.9
13	Eugenol (main compound 2)	−4.7
14	β-elemene	−4.6
15	Methyleugenol (main compound 1)	−4.3
16	(E)-caryophyllene	−4.9
17	α-humulene	−4.9
18	Germacrene D	−5.0
19	α-selinene	−5.1
20	δ-cadinene	−3.9
21	Spathulenol	−5.0
22	β-caryophyllene oxide	−5.8
23	Viridiflorol	−5.1
24	Isocembrol	−5.4

**Table 4 plants-09-01534-t004:** Phytochemicals with lowest binding energy score and interacting residues with human peroxiredoxin 5 target protein.

Compounds Receptor vs. Targets	Interacting Residues 1HD2	Binding Energy (kcal/mol)
Viridiflorol	van der Waals: Gly46, Cys47, Phe120, Arg127. H bond: Thr147 (2.86). C–H bond: Thr147 (3.28). Alkyl: Pro45 (4.75), Leu116 (4.13) (4.42) (5.43), Ile119 (5.10)	−5.1
Methyleugenol (main compound 1)	van der Waals: Ala42, Thr44, Asn76, Phe120. H bond: Gly46 (2.21) (2.41), Cys47 (2.27) (2.88). C–H bond: Pro45 (3.45). Alkyl/Pi–Alkyl: Phe43 (4.24), Val80 (4.00) (4.43)	−4.3
Isocembrol	van der Waals: Thr44, Gly46, Ile119, Arg127, Gly148. H bond: Thr147 (2.61). Alkyl/Pi–Alkyl: Pro40 (4.03), Pro45 (3.87) (4.31), Leu116 (4.50), Phe120 (5.31)	−5.4
Eugenol (main compound 2)	van der Waals: Thr44, Ile119, Phe120, Thr147. H bond: Gly46 (2.73), Arg127 (6.35). Alkyl/Pi–Alkyl: Pro45 (3.75), Leu116 (4.31) (4.68), Phe120 (4.89)	−4.7
α-selinene	van der Waals: Ala42, Thr44, Asn76, Phe120, Arg124. Pi-Sigma: Phe43 (3.80). Alkyl/Pi–Alkyl: Phe43 (5.26), Val80 (4.19) (4.56)	−5.1
β-caryophyllene oxide	van der Waals: Pro40, Thr44, Cys47, Ile119, Thr147. H bond: Gly46 (2.73), Arg127 (6.35). Alkyl/Pi–Alkyl: Pro45 (3.75), Leu116 (4.31) (4.68), Phe120 (4.89)	−5.8

**Table 5 plants-09-01534-t005:** Physicochemical properties, pharmacokinetics, drug likeness, and medicinal chemistry of six selected compounds, according to Swiss ADME software.

Entry	1	2	3	4	5	6
Pharmacokinetics/Druglikeness						
Lipinski	Yes	Yes	Yes	Yes	Yes	Yes
TPSA (Å^2^)	18.46	29.46	12.53	20.23	0.00	20.23
Consensus Log *P*_o/w_	2.58	2.25	3.68	4.75	4.40	3.42
Bioavailability Score	0.55	0.55	0.55	0.55	0.55	0.55
GI absorption	High	High	High	High	Low	High
BBB permeant	Yes	Yes	Yes	No	No	Yes
P–gp substrate	No	No	No	No	No	No
CYP1A2 inhibitor	Yes	Yes	No	No	No	No
CYP2C19 inhibitor	No	No	Yes	Yes	Yes	Yes
CYP2C9 inhibitor	No	No	Yes	Yes	Yes	No
CYP2D6 inhibitor	No	No	No	No	No	No
CYP3A4 inhibitor	No	No	No	Yes	No	No
Log *Kp* (cm/s) ^a^	−5.60	−5.69	−5.12	−4.47	−3.85	−5.00

^a^ skin permeabilit, 1: Methyleugenol, 2: eugenol, 3: β-caryophyllene oxide, 4: isocembrol, 5: α-selinene, 6: viridiflorol.

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
