# Peer review of "Antioxidant Activities of a New Chemotype of *Piper cubeba* L. Fruit Essential Oil (Methyleugenol/Eugenol): In Silico Molecular Docking and ADMET Studies"

_plants, 2020, doi:10.3390/plants9111534_

Round 1

Reviewer 1 Report

The manuscript entitled "Antioxidant activities of a new chemotype of Piper cubeba L. fruit essential oil (methyleugenol/eugenol): in silico molecular docking and ADMET studies" is based on an original research work with scientific and practical impact. This manuscript in general is well written, however there are some issues that I point out below for the improvement of the work.

  • Delete keywords that already exist in the title
  • The introduction and especially the part about the need of the work (last paragraph before the aim of the study) is very restricted. Why should expect different response of in vitro evaluation and in silico evaluation? What about information about the chemical composition of P. cubeba fruits?
  • Between lines 143-166 there are 2 references and discussion of the results. It is better to remove them in discussion section.
  • Section 4.2 about plant material should be first as 4.1.
  • The caption of Table 1 should be contain more information so to be easier for the reader to understand it

Author Response

Point-by-point response to Reviewer 1:

Dear Dr.  (Reviewer 1): Thank you for your consideration of our manuscript. We found the comments helpful, and believe our revised manuscript represents a significant improvement over our initial submission.

Open Review

English language and style

( ) Extensive editing of English language and style required
( ) Moderate English changes required
( ) English language and style are fine/minor spell check required
(x) I don't feel qualified to judge about the English language and style

Yes

Can be improved

Must be improved

Not applicable

Does the introduction provide sufficient background and include all relevant references?

( )

( )

(x)

( )

Is the research design appropriate?

(x)

( )

( )

( )

Are the methods adequately described?

( )

(x)

( )

( )

Are the results clearly presented?

( )

(x)

( )

( )

Are the conclusions supported by the results?

(x)

( )

( )

( )

Comments and Suggestions for Authors

The manuscript entitled "Antioxidant activities of a new chemotype of Piper cubeba L. fruit essential oil (methyleugenol/eugenol): in silico molecular docking and ADMET studies" is based on an original research work with scientific and practical impact. This manuscript in general is well written, however there are some issues that I point out below for the improvement of the work.

  • Delete keywords that already exist in the title

Done

  • The introduction and especially the part about the need of the work (last paragraph before the aim of the study) is very restricted. Why should expect different response of in vitro evaluation and in silico evaluation? What about information about the chemical composition of P. cubeba fruits?

This paragraph was modified as follows:

This work was conducted due to the lack studies on the chemical composition of P. cubeba fruit’s essential oil in parallel with the search for new powerful chemotype as well as the originality ……

Also the following paragraph was added to discussion part:

On the other hand, the chemical composition of P. cubeba fruit’s essential oil was scarcely explored. Previous reports showed that the main constituents are were β-elemene (9.4%) and sabinene (9.1%) [34]. In another study by Burfield [35], β-caryophyllene, δ-cadinene, α- and β-cubebenes, and minor amounts of monoterpenes were found as the main phytocompounds Lawrence [36] reported that the main components were α-copaene, β-cubebene, allo-aromadendrene, γ-muurolene and germacrene D followed by δ-cadinene and β-caryophyllene. In the recent study of Andriana et al. [37], the authors found that terpinen-4-ol (42.41%), α-copaene (20.04%), and γ-elemene (17.68%) were the major components. Based on the above literature descriptions, our chemical composition results are still different and confirm the originality of our P. cubeba fruit’s essential oil.

[34] Bos, R.; Woerdenbag, H.J.; Kayser, O.; Quax,W.J.; Ruslan, K.; Elfami. Essential oil constituents of Piper cubeba L. fils. from Indonesia. J. Essent. Oil Res. 2007, 19, 14–17.

[35] Burfield, T. Natural Aromatic Materials: Odours & Origins, 2nd ed.; The Atlantic Institute of Aromatherapy: Tampa, FL, USA, 2017.

 [36] Lawrence, B.M. Progress in essential oils: Cubeb oil. Perfum. Flavor. 2016, 41, 54–57.

[37] Andriana, Y.; Xuan T.D. ; Quy T.N. ; Tran H.D. ; Le Q.T. Biological Activities and Chemical Constituents of Essential Oils from Piper cubeba Bojer and Piper nigrum L. Molecules 2019, 24, 1876; doi:10.3390/molecules24101876.

  • Between lines 143-166 there are 2 references and discussion of the results. It is better to remove them in discussion section.

Now this paragraph has been moved

  • Section 4.2 about plant material should be first as 4.1.

This remark has been taken into account

  • The caption of Table 1 should be contain more information so to be easier for the reader to understand it

Now, the legend of Table 1 has been modified as suggested by the reviewer

Reviewer 2 Report

The manuscript is interesting, well structured and with conclusions in line with the results achieved. However, the authors should / must: - indicate correctly the names and formulas of the ROS, - indicate correctly the structures of the identified compounds (the wrong structure of one is reported), highlighting the correct stereochemistry, - correct a series of mistakes highlighted in yellow in the attached file, - report the references according to the journal specifications

Author Response

Point-by-point response to Reviewer 2:

Dear Dr.  (Reviewer 2): Thank you for your consideration of our manuscript. We found the comments helpful, and believe our revised manuscript represents a significant improvement over our initial submission.

Open Review

English language and style

( ) Extensive editing of English language and style required
( ) Moderate English changes required
(x) English language and style are fine/minor spell check required
( ) I don't feel qualified to judge about the English language and style

Yes

Can be improved

Must be improved

Not applicable

Does the introduction provide sufficient background and include all relevant references?

( )

(x)

( )

( )

Is the research design appropriate?

(x)

( )

( )

( )

Are the methods adequately described?

(x)

( )

( )

( )

Are the results clearly presented?

( )

(x)

( )

( )

Are the conclusions supported by the results?

(x)

( )

( )

( )

Comments and Suggestions for Authors

The manuscript is interesting, well structured and with conclusions in line with the results achieved. However, the authors should / must: - indicate correctly the names and formulas of the ROS, - indicate correctly the structures of the identified compounds (the wrong structure of one is reported), highlighting the correct stereochemistry, - correct a series of mistakes highlighted in yellow in the attached file, - report the references according to the journal specifications

  • indicate correctly the names and formulas of the ROS,

…….the formation of reactive oxygen/nitrogen species (ROS/RNS) such as superoxide anion (Ȯ2-•), peroxide radical (•OOH), singlet oxygen (1O2), hydroxyl radical (•OH), nitric oxide (NO•) and peroxynitrite (ONOO).

  • indicate correctly the structures of the identified compounds (the wrong structure of one is reported),

All structures have been verified (see Figure 2)

  • correct a series of mistakes highlighted in yellow in the attached file

Now they have been corrected as indicated by the reviewer.

  • report the references according to the journal specifications
    All references have been checked
